# A Cross-Sectional Study on the Relationship between the Family Nutrition Climate and Children’s Nutrition Behavior

**DOI:** 10.3390/nu11102344

**Published:** 2019-10-02

**Authors:** Sacha Verjans-Janssen, Dave Van Kann, Stef Kremers, Steven Vos, Maria Jansen, Sanne Gerards

**Affiliations:** 1Department of Health Promotion, NUTRIM School of Nutrition and Translational Research in Metabolism, Maastricht University, 6229 HA Maastricht, The Netherlands; 2School of Sport Studies, Fontys University of Applied Sciences, 5644 HZ Eindhoven, The Netherlands; 3Department of Industrial Design, Eindhoven University of Technology, 5612 AZ Eindhoven, The Netherlands; 4Academic Collaborative Center for Public Health, Public Health Service South-Limburg, 6400 AA Heerlen, The Netherlands; 5Department of Health Services Research, Maastricht University, CAPHRI Care and Public Health Research Institute, 6229 GT Maastricht, The Netherlands

**Keywords:** family, parents, health, nutrition, children

## Abstract

Background: Parents influence their children’s nutrition behavior. The relationship between parental influences and children’s nutrition behavior is often studied with a focus on the dyadic interaction between the parent and the child. However, parents and children are part of a broader system: the family. We investigated the relationship between the family nutrition climate (FNC), a family-level concept, and children’s nutrition behavior. Methods: Parents of primary school-aged children (*N* = 229) filled in the validated family nutrition climate (FNC) scale. This scale measures the families’ view on the consumption of healthy nutrition, consisting of four different concepts: value, communication, cohesion, and consensus. Parents also reported their children’s nutrition behavior (i.e., fruit, vegetable, water, candy, savory snack, and soda consumption). Multivariate linear regression analyses, correcting for potential confounders, were used to assess the relationship between the FNC scale (FNC-Total; model 1) and the different FNC subscales (model 2) and the child’s nutrition behavior. Results: FNC-Total was positively related to fruit and vegetable intake and negatively related to soda consumption. FNC-value was a significant predictor of vegetable (positive) and candy intake (negative), and FNC-communication was a significant predictor of soda consumption (negative). FNC-communication, FNC-cohesion, and FNC-consensus were significant predictors (positive, positive, and negative, respectively) of water consumption. Conclusions: The FNC is related to children’s nutrition behavior and especially to the consumption of healthy nutrition. These results imply the importance of taking the family-level influence into account when studying the influence of parents on children’s nutrition behavior. Trial registration: Dutch Trial Register NTR6716 (registration date 27 June 2017, retrospectively registered), METC163027, NL58554.068.16, Fonds NutsOhra project number 101.253.

## 1. Introduction

Worldwide, children’s body mass index (BMI) and the prevalence of overweight and obesity among children increased in the last 40 years [1]. Although the trend concerning the age-standardized BMI of children in the northwestern part of Europe is flattening, numbers remain high. In 2018, almost 12% of Dutch children between the ages of four and 11 years were overweight or obese [2]. Overweight and obesity in childhood are associated with physical and psychological morbidity in the short and long term [3]. A cause of overweight and obesity is high-energy intake, in the form of energy-dense, nutrient-poor foods and soft drinks [4]. Many children consume too much sugar, e.g., sugar-containing beverages and energy-dense snacks [5], and consume insufficient amounts of healthy foods, such as fruits and vegetables [4]. Healthy eating behaviors in childhood decrease the risk of health-related diseases at a later age, and are related to having healthy eating habits as an adult [6,7].

During childhood, parents decide what children eat, gradually shaping the child’s nutritional behavior. Parents do this by exhibiting specific, goal-directed behaviors, i.e., food parenting practices [8,9]. Certain food parenting practices, such as controlling the availability of healthy or unhealthy foods, and parental modeling of eating behaviors, are most consistently related to children’s nutrition behavior [10,11]. On the other hand, other food parenting practices, such as pressuring to eat, monitoring, and rewarding food consumption, are not consistently related to children’s nutrition behavior [10,11]. This is probably due to the influence of general parenting on these parent–child interactions [8,11,12,13]. General parenting refers to the emotional climate in which the parenting practices are expressed [8]. Children show a healthier nutrition behavior as a result of certain parenting practices (e.g., encouragement) when the parents provide a positive parenting climate [13]. A positive parenting climate is characterized by nurturance, structure, and behavioral control [14].

Many studies on the relationship between food parenting practices, general parenting, and children’s nutrition behavior assume a unidirectional communication in which the child is the mere recipient [15]. However, in reality, parents and children are part of a family in which family members influence each other’s behaviors, indicating a reciprocal influence [16]. Focusing only on individual parenting practices and general parenting, and not considering this family-level influence can lead to important information being missed when studying the parents’ influence on children’s nutrition behavior [15,17].

Very little research was conducted on the family-level influence on children’s nutrition behavior. It is hypothesized that the family’s emphasis on healthy nutrition and their interactions concerning nutrition behavior (e.g., parents discussing whether it is important to eat healthy as a family, consuming meals together as a family) influence whether and how much healthy nutrition is consumed by the child. These familial interactions and views are part of a family’s health climate. To capture this family climate concerning healthy nutrition, Niermann and colleagues [18] developed the family nutrition climate (FNC) scale (as part of the family health climate scale). The FNC is considered “the [family’s] shared perceptions and cognitions” concerning healthy nutrition behavior on a daily basis [18] (p. 31).

The FNC is a relatively new construct and, to our knowledge, only two studies were conducted using this validated instrument to measure the relationship between the family and individual nutrition behavior [19,20]. Both studies found a relationship between the FNC and individual nutrition behavior: families with healthy behavioral patterns (consuming high levels of healthy nutrition and showing high levels of physical activity) rated their FNC higher compared to families with unhealthier behavioral patterns [20]. In the second study, it was found that a higher score on the FNC was related to a higher consumption of children’s healthy foods (i.e., fruit, vegetables, and salad) [19]. In both studies, the children were adolescents with a mean age of 14 years. It is unknown whether these results are similar for a population of younger children. Therefore, this study aimed to investigate the relationship between the FNC and the nutrition behavior of Dutch children aged 7–10 years old.

## 2. Methods

### 2.1. Study Design and Procedure

For the current study, baseline data of a quasi-experimental study were used. The protocol of this quasi-experimental intervention study was described in Verjans-Janssen et al. [21]. Ethical approval for the study was provided by the Medical Ethics Committee of the Maastricht University Medical Centre and Maastricht University (METC163027, national number: NL58554.068.16). After being informed about the study, all participants consented to participate before the start of the study. The parents of the participants signed an informed consent form. The questionnaires are available in Dutch from the corresponding author upon request.

Primary school children in grades 4–6 (Dutch educational system: between the ages of seven and 10 years) were eligible for inclusion. Eleven primary schools participated in the study. The schools were located in low socio-economic neighborhoods in two cities in the southern region of the Netherlands.

The children were provided with oral and written information by a researcher. Their parents received written information about the study and were given the opportunity to ask questions during scheduled information hours or via phone or e-mail. The children were included in the study if both parents provided written consent. Of the eligible children, 60.4% consented to participate in the study.

### 2.2. Measurements

Data were collected in March and April 2017. Data collection tools included a child questionnaire, a parent questionnaire (both paper-based), and anthropometric measurements. The child questionnaire was filled in on paper in the classroom during school hours. The child’s anthropometric measurements were taken by trained researchers at school. Parents filled in the parent questionnaire at home and returned the questionnaire via the child to school. The questionnaires are available in Dutch from the corresponding author upon request.

#### 2.2.1. Children’s Nutrition Behavior

The child’s nutrition behavior was measured with the parent questionnaire. For this, items of a validated food frequency questionnaire for Dutch children was used [22]. Children’s fruit, vegetable, water, candy (i.e., sweets, licorice, candy bars), savory snack (e.g., cheese, crisps), and soda intake were measured. For water intake, the consumption of tea without sugar was also included. For soda, only sugar-containing drinks were included (diet soda sweetened with artificial sweeteners was excluded from the analysis). The frequency (from zero to seven days a week) and the amount per day of these foods and drinks were measured. The amount was measured in natural units: pieces for fruit, serving spoons for vegetables, portions for candy (i.e., a handful of sweets or one normal-sized candy bar), portions for savory snacks (i.e., a handful of cheese or a bowl of crisps), and glasses for water and soda. The frequency and the amount were multiplied and divided by seven to calculate an average daily consumption.

#### 2.2.2. Family Health Climate

The FNC scale is a validated questionnaire [18], which was translated into Dutch [17]. It can be completed by one family member or by multiple family members, after which an aggregated FNC score can be calculated [18,19]. In this study, one parent filled out the FNC. The higher the score is on the FNC scale, the more importance is placed on eating healthy as a family [18]. The FNC scale consists of four subscales: FNC-value, FNC-communication, FNC-cohesion, and FNC-consensus. FNC-value (four items) encompasses the family’s interest in healthy nutrition in daily life and whether healthy nutrition is a norm within the family (e.g., “In our family, a healthy diet plays an important role in our lives”) (α = 0.87). FNC-communication (five items) comprises the family’s talks about healthy nutrition and support regarding eating healthy foods (e.g., “In our family, we are interested in articles (e.g., magazines) on healthful nutrition”) (α = 0.81). FNC-cohesion (five items) encompasses whether consuming meals together as a family is common and whether eating together is considered important (e.g., “In our family, we appreciate spending time together during meals”) (α = 0.67). FNC-consensus (three items) refers to the agreement within a family concerning nutrition (e.g., “In our family, we rarely argue about food- or diet-related matters”) (α = 0.80). All items were assessed on a four-point Likert scale (“strongly disagree” to “strongly agree”). The average of the four FNC-subscale scores was taken to obtain the FNC overall score (FNC-Total; 17 items) (α = 0.86). Missing data for the FNC scale were imputed when there was 10% or less of missing items. A missing value was replaced by the mean of the available FNC subscale items for that particular participant. If a subscale had only one valid item, data were not imputed, and the participant was excluded from analysis. Imputation on the FNC scale was applied for nine participants.

#### 2.2.3. Children’s Weight Status

The children were weighed and measured during the schools’ physical education lessons. They wore light sports clothes during the anthropometric measurements, but shoes were taken off. Weight was measured to the nearest 0.1 kg with a digital weighing scale (Seca 803), and height was measured to the nearest 0.1 cm with a portable stadiometer (Seca 213). For the children with missing data due to absence on the day of measurement, data were imputed from the parent questionnaire (i.e., self-reported data); this was the case for five children. Children’s BMI *z*-scores were calculated using a Dutch reference population and adjusting for the child’s age and gender [23].

#### 2.2.4. Socio-Demographic Characteristics

Socio-demographic characteristics of the child were measured with the child questionnaire. Children reported their date of birth, their gender, and the number of brothers and sisters. This was used to calculate the number of children within a family. Children reported the birth country of both parents, which was used to define the children’s ethnicity. Ethnicity was based on the definition of Western and non-Western by Statistics Netherlands [24]; the child’s ethnicity was non-Western when at least one parent was born in a non-Western country.

In the parent questionnaire, the parents reported on their educational level, parent relationship (married/having a partner or single), and weight and height. Educational level was recoded into three categories based on the International Standard Classification of Education 2011 [25]: no or primary educational level (no education or primary school), secondary educational level (pre-vocational school, secondary education, or lower vocational education), and tertiary educational level (higher vocational education or university degree). Parent’s body mass index (BMI) was calculated and used to define the weight status of the parent (i.e., BMI <20: underweight, BMI 20–25: normal weight, BMI 25–30: overweight, and BMI >30: obese).

### 2.3. Participants

In total, 523 children participated in the study. Only data of parent–child dyads were used. Of the 523 children, 329 parents (62.9%) provided data on their child(ren)’s nutrition behavior and the family nutrition climate. To ensure that independent samples were included, the data of one sibling of a sibling pair were excluded; the sibling with the lowest birth month was excluded. In the case of twins, the child who was first in the dataset (sorted by subject identifier) was excluded. In total, 18 siblings were excluded. Only cases with complete data were included in the analyses.

### 2.4. Statistical Analysis

Data were analyzed with SPSS version 24.0 (IBM Corp., Armonk, NY, USA). Descriptive statistics were used to describe the characteristics of the study population and the mean values and standard deviations of FNC-Total, the FNC subscales, and the nutrition behavior outcomes. Differences in the FNC between different types of families, based on the child’s ethnicity (Western versus non-Western) and the parent’s education level (low versus high), were analyzed. As the FNC data were not normally distributed, differences between these groups were analyzed with a Mann–Whitney U-Test.

Data of the nutrition behaviors were square-root-transformed due to violation of the normality assumption of the residuals in the regression analyses. Pearson correlation analyses were conducted to obtain the correlations between the FNC-Total/FNC subscales and nutrition behavior outcomes. Multivariate linear regression analyses were used to examine the relationship between the FNC and the individual nutrition behavior outcomes, while correcting for potential confounders: age, gender, ethnicity, and BMI *z*-score of the child, and parent’s educational level, BMI, and age. These covariates were entered simultaneously into the linear regression models.

Two types of linear regression models were built. Firstly, regression models were built for all foods and drinks, separately, with FNC-Total as an independent variable (model 1). Secondly, the regression models were restructured by replacing FNC-Total with the FNC subscales (i.e., FNC-value, FNC-communication, FNC-cohesions, FNC-consensus), which were added simultaneously as independent variables to the models (model 2). A two-sided *p*-value of less than 0.05 was considered statistically significant.

## 3. Results

### 3.1. Characteristics of the Study Population

In total, 229 (69.6%) parent–child dyads were included in the analyses. The mean age of the children was 8.3 years (SD = 1.0). More girls (62.0%) than boys participated in the study. Mainly mothers filled in the questionnaire (83.8%). The mean age of the parents was 38.6 years (SD = 5.4). Of the parents, 35.4% had a secondary educational level, and 38.9% had a tertiary educational level (higher vocational degree or higher). Of the remaining parents, 22.7% finished primary education, and 3.1% (*N* = 7) reported having no education. Of all children, slightly more than one-quarter (27.1%) had a non-Western ethnicity, with one or both parents born in a non-Western country. Overall, 17.9% of the children and 42.8% of the parents were overweight and obese. Regarding the family situation, the majority of the parents had a partner (81.7%) and two or more children (83.4%) (Table 1).

### 3.2. Means of the Child’s Nutrition Behavior and the Family Nutrition Climate

Children consumed on average 1.4 pieces of fruit (SD = 0.6), 1.8 portions (±90 g) of vegetables (SD = 0.9), and 2.3 glasses of water daily. On average, they consumed a little less than one portion of candy (mean (M) = 0.9, SD = 0.7), a little less than half a portion of savory snacks (M = 0.4, SD = 0.4), and 1.4 glasses of soda (SD = 1.3) daily (Table 2).

The average scores on the FNC subscales were high (>3), except for FNC-communication (M = 2.9, SD = 0.6), see Table 2. Parents scored highest on FNC-cohesion (M = 3.7, SD = 0.4).

FNC-Total did not differ significantly for families of different ethnicity (U = 4735.0, *p* = 0.32), but it did differ significantly between less well-educated and highly educated parents (U = 4121.5, *p* = 0.04). Highly educated parents scored higher on the FNC (M = 3.3, SD = 0.4) compared to less well-educated parents (M = 3.2, SD = 0.4).

### 3.3. Relationship between the Family Nutrition Climate and the Child’s Nutrition Behavior

FNC-Total correlated positively with healthy food consumption, i.e., fruit and vegetable intake, and negatively with unhealthy food and drink consumption, i.e., candy intake and soda consumption (Table 2). FNC-value correlated with the same food and drink consumption as FNC-Total, and in the same direction, i.e., positively with fruit and vegetable intake and negatively with candy intake and soda consumption. FNC-communication correlated positively with all healthy foods (fruit intake, vegetable intake, and water consumption) and negatively with soda consumption. There were no significant correlations between FNC-cohesion and nutrition behavior or between FNC-consensus and nutrition behavior.

Adjusting for child and parent characteristics, FNC-Total was a significant predictor of fruit intake (standardized β = 0.15), vegetable intake (β = 0.23), and soda consumption (β = −0.20) (Table 3). FNC-value was positively related to vegetable intake (β = 0.34) and negatively to candy consumption (β = −0.19). FNC-communication was positively related to water consumption (β = 0.19) and negatively to soda consumption (β = −0.24). Regarding water consumption, FNC-cohesion and FNC-consensus were also predictors, but in opposite directions (β = 0.15 and β = −0.20, respectively).

## 4. Discussion

The current study investigated the relationship between the FNC and the nutrition behavior of primary school-aged children. We found that FNC-Total was a positive predictor of the consumption of healthy nutrition (i.e., fruit and vegetable intake) and a negative predictor of the consumption of one unhealthy nutrition (i.e., soda consumption). The results were in line with our expectation that the FNC is mainly a predictor of children’s healthy nutrition consumption. Our results show that a family-level influence is present on children’s nutrition behavior. Other research already showed that family members influence each other’s nutrition behaviors. For example, families have similar dietary intakes [26,27], especially regarding the consumption of healthy nutrition [26]. Our study specifically shows the influence of shared values, routines, and interaction patterns concerning healthy nutrition within a family (defined as the family nutrition climate) on the child’s nutrition behavior.

In this study, the parents rated FNC-cohesion the highest, indicating that consuming meals together as a family was common and considered important. Family meals are related to a healthier nutrition behavior of children [28]. However, in our study, we only found a significant association between FNC-cohesion and water consumption. It can be that, although family meals were common, these moments did not consist of interpersonal contact. Research showed that distractions during family meals, such as having the television on, is negatively related to group enjoyment during the meals (even when the family is not paying attention to the television) [29]. The quality of the time spent together during family meals, e.g., talking about each other’s day and discussing the importance of healthy nutrition during family meals, is likely a more important part of the family nutrition climate and of more influence on children’s nutrition behavior than merely sitting together at the dinner table. In our study, we indeed found a stronger association between the communication within the family concerning healthy nutrition and FNC-Total than between FNC-cohesion and FNC-Total.

FNC-Total was associated the strongest with children’s vegetable consumption. In the Netherlands, children consume vegetables mainly during dinner in the form of cooked vegetables [30,31]. Dinner is most of the times consumed at home with the family and more regularly compared to breakfast and lunch [32,33], which explains the strong association between FNC-Total and children’s vegetable consumption. Research showed that parents highly value the consumption of vegetables by their child because of the health benefits [34]. Our results showed that FNC-value was the only significant predictor of the child’s vegetable intake.

While vegetables are mainly consumed during meals, fruit, candy, and snacks are mainly consumed in between meals [5,31]. However, this does not imply that the family influence is not present regarding the consumption of fruit. On the contrary, FNC-total was also related to the child’s fruit consumption, but to a lesser extent than vegetable consumption. This can be explained by the fact that Dutch primary school-aged children consume at least 70% of their fruit intake at home, and about 20% at school [31]. Since the schools in this study participated in the “European Union (EU) school fruit” program, in which children were provided with fruit at least three days a week during morning recess [35], there was also an influence of the school environment on the children’s fruit consumption.

The third association between FNC-Total and children’s nutrition behavior involved children’s soda consumption. The FNC refers to the importance of a healthy diet within the family. Although soda is not a healthy food, the FNC was associated with soda consumption; however, this was in the preferred direction (i.e., lower soda intake). This desirable family influence on soda intake was present because Dutch primary school children consume soda mainly at home. Furthermore, 11.3% of the daily soda consumption takes place during dinner [5], which is mainly consumed at home and with the family, as mentioned earlier. Our results showed that the family influence on soda consumption was present in the communication within the family concerning healthy nutrition (FNC-communication). Based on these results and the positive relationship between FNC-communication and water consumption, we hypothesize that the family supports each other in refraining from soda and talks about water as being a healthy drink.

This positive relationship between FNC-communication and children’s water consumption and between FNC-cohesion and children’s water consumption countered the negative relationship between FNC-consensus and children’s water consumption. These contrasting results explain the non-significant relationship between FNC-Total and children’s water consumption. Although a higher score on FNC-consensus was assumed to positively relate to a healthier nutrition behavior, the negative association between FNC-consensus and water consumption might be explained by the framing of the items, i.e., “In our family, we rarely argue about food- or diet-related matters”. The unexpected relationship could either imply that there is no arguing about drinking water or that water consumption is insufficiently considered to be part of “food- or diet-related matters”.

There was no association between FNC-Total and candy and savory snacks. It may be that the family influence on the consumption of these foods is less present compared to the other foods. One-third of the time, Dutch primary school-aged children consume candy and snacks at a friend’s place or when outside [5] and, thus, away from the family environment. Another explanation may be that snacks and candies are not healthy foods and are, thus, not taken into consideration when thinking about consuming healthy nutrition within the family.

Our results are in line with the study on the FNC and adolescents’ nutrition behavior [19]. Niermann and colleagues [19] found a positive association between the aggregated FNC and adolescents’ healthy dietary behavior, consisting of the intake of salads, vegetables, and fruits. The effect size of their association was medium. The associations found in the current study were less strong, while the influence of the family on the nutrition behavior of younger children was expected to be stronger. A likely explanation for the stronger associations found in the study by Niermann et al. [19] is that they included more highly educated parents (41.9% of the mothers and 58.4% of the fathers). In our study, highly educated parents had a significantly higher FNC-Total score compared to less well-educated parents. Additional interaction analysis was conducted to see whether educational level had an influence on the relation between the FNC and the children’s nutrition behavior, but no significant interaction between FNC-Total and the parent’s educational level was found (data not shown). However, our sample was rather small, and further research is required to investigate whether different types of families (e.g., based on ethnicity, educational level, socioeconomic status) differ in the FNC and how this difference is related to the children’s nutrition behavior.

The family can be considered a social dynamic system consisting of different subsystems (e.g., individuals, spouse subsystem, parent–child subsystem) that interact with each other. To understand a family’s properties and their influence, the different parts of a family cannot simply be combined because they are interdependent (e.g., family functioning depends on an interplay of communication patterns and role fulfillment) [15,36]. Given the complexity of the family influences on the children’s energy balance-related behaviors, it is quite challenging to study this family system [15]. The FNC is a part of this complex family system and, by the use of the FNC-scale, we aimed to capture some of the influence of this complex family system on children’s nutrition behavior.

The rather weak associations found between the FNC and the children’s nutrition behaviors in this study can be explained by the fact that the FNC operates on a more distal level within the family system [15]. However, these weak associations do not imply that this family-level influence is less relevant. On the contrary, the FNC is an important part of the family system, which is associated with the children’s nutrition behavior, as shown in our study, via mediated paths (e.g., through intrinsic motivation), as shown in the study of Niermann et al. [19], and as a higher-level moderator (i.e., stronger relationships were found between food parenting practices and children’s BMI *z*-score when the family nutrition climate was healthier), as shown in the study of Gerards et al. [17].

### 4.1. Strengths and Limitations

To our knowledge, this is the first study to investigate the relationship between the FNC and the nutrition behavior of primary school-aged children. The strengths of this study were the diversity of the sample regarding the children’s ethnicity and the educational level of the parents.

A limitation of the study was the underrepresentation of paternal views on the FNC. Paternal views on FNC may differ from maternal views [37]. The underrepresentation of fathers in studies is a common limitation of observational studies on parenting and children’s obesity-related behaviors, especially nutrition behavior [38]. This possibly limits the generalizability of the results.

Another limitation possibly affecting generalizability of the results might be the slight overrepresentation of girls in the sample. However, in the study on the association between the FNC and adolescents’ consumption of healthy foods, the correlations did not differ significantly between boys and girls [19].

Other limitations were the cross-sectional design and the use of a self-reporting instrument to measure the children’s nutrition behavior. Longitudinal studies should be conducted between the FNC and the children’s energy balance-related behaviors to study the stability of the FNC over time and how the FNC influences the children’s energy balance-related behaviors over time. We also recommend objective measurements of the children’s nutrition behavior, such as the use of wearables [39], because self-reporting instruments for nutrition behavior are prone to social desirability.

Finally, it is debatable whether our choice of nutrition behaviors is the best representation of children’s healthy and unhealthy nutrition behaviors, since other behaviors such as breakfast consumption or other foods and drinks such as milk are also part of the children’s daily nutrition behavior.

### 4.2. Implications and Recommendations

Our results add to the existing knowledge of the family influence on children’s nutrition behavior. The results underline the importance of addressing the whole family system instead of focusing merely on the parent–child subsystem [15,40]. Addressing the family environment should be done by involving all family members [19]. The healthy family environment can be assessed by the individual evaluations of the FNC. Ideally, the FNC is measured by assessing the views of all family members (the parents, the child, and siblings) [18].

To inform intervention developers, we recommend further research into the interaction between the more proximal parent-level influences (e.g., parenting practices) and the more distal family-level factors (e.g., the FNC) and their interacting influence on the children’s nutrition behavior. To be able to study these interacting influences, we advocate studying general parental influences, i.e., general parenting, and more specific parenting, i.e., nutrition parenting practices, as well as the broader family context, i.e., the family nutrition climate.

To measure general parenting, we recommend the use of the validated comprehensive general parenting questionnaire for caregivers of 5–13-year-old children of Sleddens et al. [41]. This questionnaire was developed after a thorough search of the literature and assesses five parenting constructs: nurturance, structure, behavioral control, overprotection, and coercive control [41]. Unfortunately, there is little consensus on how to best measure food parenting practices [42]. There are at least 71 instruments measuring food parenting practices [43]. In most cases, these instruments measure a small part of the spectrum of food parenting practices [43]. Fortunately, a comprehensive food parenting practices item bank is currently being developed, which will allow a more consistent and comprehensive measurement of food parenting practices in the future [42]. For the measurement of the family-level influence, we recommend the use of the FNC.

## 5. Conclusions

The climate within a family concerning healthy nutrition (e.g., valuing healthy nutrition within a family, and communicating about eating healthy as a family) is a predictor of the children’s nutrition behavior, especially the consumption of healthy foods. These results indicate the importance of considering family-level influences when aiming to improve children’s nutrition behavior.

## Figures and Tables

**Table 1 nutrients-11-02344-t001:** Characteristics of the target population (*N* = 229).

	Number	%	Mean	SD
*Parent characteristics*				
Parent				
Mother	192	83.8		
Father	37	16.2		
Age			38.6	5.4
Educational level				
No or primary education	59	25.8		
Secondary education	81	35.4		
Tertiary education	89	38.9		
Weight status				
Underweight or normal weight	131	57.2		
Overweight or obese	98	42.8		
*Child characteristics*				
Age			8.3	1.0
Gender				
Male	87	38.0		
Female	142	62.0		
Ethnicity				
Western	167	72.9		
Non-Western	62	27.1		
Weight status				
Underweight or normal weight	188	82.1		
Overweight or obese	41	17.9		
*Family situation*				
Relationship				
Living together with a partner	187	81.7		
Single	42	18.3		
Number of children				
1 child	38	16.6		
2 or more children	191	83.4		

**Table 2 nutrients-11-02344-t002:** Correlation coefficients between family nutrition climate and the child’s individual nutrition behavior.

			Pearson Correlation Coefficients									
			1	2	3	4	5	6	7	8	9	10
	*Family Nutrition Climate*	M (SD)										
1	Total	3.3 (0.4)	-	-	-	-	-	-	-	-		
2	Value	3.4 (0.6)	**0.83**	-	-	-	-	-	-	-		
3	Communication	2.9 (0.6)	**0.81**	**0.59**	-	-	-	-	-	-		
4	Cohesion	3.7 (0.4)	**0.56**	**0.33**	**0.21**		-	-	-	-		
5	Consensus	3.1 (0.7)	**0.68**	**0.46**	**0.30**	**0.33**	-	-	-	-		
	*Individual nutrition behavior ^a^*											
6	Fruit (pieces/day)	1.4 (0.6)	**0.18**	**0.16**	**0.21**	0.08	0.01	-	-	-		
7	Vegetables (portions/day) ^b^	1.8 (0.9)	**0.26**	**0.32**	**0.19**	0.10	0.12	0.12	-	-		
8	Water (glasses/day)	2.3 (1.7)	0.10	0.07	**0.17**	0.11	−0.10	**0.17**	**0.23**	-		
9	Candy (portions/day) ^c^	0.9 (0.7)	**−0.15**	**−0.19**	−0.09	−0.12	−0.05	0.06	−0.11	0.02		
10	Snacks (portions/day) ^d^	0.4 (0.4)	−0.05	−0.08	0.01	−0.04	−0.06	**0.17**	−0.06	−0.07	**0.30**	
11	Soda (glasses/day)	1.4 (1.3)	**−0.23**	**−0.21**	**−0.29**	−0.04	−0.07	−0.07	**−0.14**	**−0.39**	0.11	0.12

Note: M: mean, SD: standard deviation; bold numbers are statistically significant (*p* < 0.05). ^a^ The data for the nutrition behaviors were transformed (square root). ^b^ One serving spoon is approximately 50 g. ^c^ One portion is a handful of sweets or a normal-sized candy bar, for example. ^d^ One portion is a handful of cheese or a bowl of crisps, for example.

**Table 3 nutrients-11-02344-t003:** Associations between the family nutrition climate (FNC) and the child’s individual nutrition behavior (linear regression models).

		Model 1			Model 2		
	FNC	Unst. B	(SE)	St. β	Unst. B	(SE)	St. β
*Individual nutrition behavior ^a^*							
Fruit	*FNC-Total*	**0.11**	**(0.05)**	**0.15**			
(pieces/day)	Value				0.07	(0.10)	0.06
	Communication				0.15	(0.08)	0.15
	Cohesion				0.15	(0.13)	0.08
	Consensus				−0.10	(0.07)	−0.10
	*R^2^*			0.12			0.15
Vegetables	*FNC-Total*	**0.19**	**(0.05)**	**0.23**			
(serving spoons/day) ^b^	Value				**0.20**	**(0.05)**	**0.34**
	Communication				−0.02	(0.04)	−0.04
	Cohesion				−0.01	(0.07)	−0.01
	Consensus				0.00	(0.04)	0.00
	*R^2^*			0.12			0.16
Water	*FNC-Total*	0.13	(0.10)	0.09			
(glasses/day)	Value				−0.03	(0.10)	−0.03
	Communication				**0.18**	**(0.08)**	**0.19**
	Cohesion				**0.26**	**(0.12)**	**0.15**
	Consensus				**−0.17**	**(0.07)**	**−0.20**
	*R^2^*			0.06			0.11
Candy	*FNC-Total*	−0.10	(0.06)	−0.12			
(portions/day) ^c^	Value				**−0.11**	**(0.06)**	**−0.19**
	Communication				0.03	(0.04)	0.05
	Cohesion				−0.09	(0.07)	−0.09
	Consensus				0.03	(0.04)	0.06
	*R^2^*			0.07			0.09
Snacks	*FNC-Total*	−0.04	(0.13)	−0.02			
(portions/day) ^d^	Value				−0.07	(0.12)	−0.06
	Communication				0.13	(0.10)	0.11
	Cohesion				−0.04	(0.16)	−0.02
	Consensus				−0.09	(0.09)	−0.08
	*R^2^*			0.05			0.06
Soda	*FNC-Total*	**−0.31**	**(0.10)**	**−0.20**			
(glasses/day)	Value				−0.06	(0.10)	−0.06
	Communication				**−0.23**	**(0.08)**	**−0.24**
	Cohesion				0.04	(0.13)	0.02
	Consensus				0.02	(0.07)	0.02
	*R^2^*			0.13			0.15

Note: Model 1: FNC-Total as independent variable. Model 2: FNC-value, FNC-communication, FNC-cohesions, and FNC-consensus as independent variables. All models were adjusted for child’s age, gender, body mass index (BMI) *z*-score, and ethnicity, and parent’s educational level, BMI, and age. Bold numbers are significant, *p* < 0.05. Unst. = unstandardized, St. = standardized. ^a^ The data for the nutrition behaviors were transformed (square root). ^b^ One serving spoon is approximately 50 g. ^c^ One portion is a handful of sweets or a normal-sized candy bar, for example. ^d^ One portion is a handful of cheese or a bowl of crisps, for example.

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
