# Peer review of "A Cross-Sectional Study on the Relationship between the Family Nutrition Climate and Children’s Nutrition Behavior"

_nutrients, 2019, doi:10.3390/nu11102344_

Round 1
Reviewer 1 Report
The paper is well designed and important for child health and thus the health of the next generation.
English language editing can be done in-house.
one concern that must be discussed in the text: the skewed response rate with majority of mothers and more than 60 % girls included. That must be commented, because we might miss some of the real problems.
Author Response
Reviewer 1
Dear reviewer,
Thank you for your compliments on the manuscript and the suggestions to improve the manuscript. Below we will provide a point-by-point response to your comments.
Reviewer comment 1:
“English language editing can be done in-house.”
Author’s response, comment 1:
The manuscript was sent to a professional editor (a native English speaker) before submitting the manuscript. However, we agree that some further changes could be made to improve the English and readability of the manuscript. Therefore, we read the manuscript thoroughly and corrected spelling mistakes. In addition, we rewrote some parts of the text to improve the readability of the manuscript. Our changes are marked in the manuscript. The main changes were done in the discussion (line 250 – 357). Besides, we decided to talk about food and drink consumption or nutrition behavior instead of nutrition behaviors. Example of a correction made:
Results, line 224: FNC-Total correlated positively with healthy food consumption, i.e., fruit and vegetable intake, and negatively with unhealthy food and drink consumption, i.e., candy intake and soda consumption (Table 2).
Reviewer comment 2:
“one concern that must be discussed in the text: the skewed response rate with majority of mothers and more than 60 % girls included. That must be commented, because we might miss some of the real problems.”
Author’s response, comment 2:
Thank you for the suggestion regarding discussing the overrepresentation of mothers and girls in the study population. You have make a valid point. This overrepresentation may affect the generalizability of the results. We decided to discuss this in the ‘Limitations and strengths’-section.
Corrections:
Discussion, line 362-369: A limitation of the study was the underrepresentation of paternal views on the FNC. Paternal views on FNC may differ from maternal views [37]. The underrepresentation of fathers in studies is a common limitation of observational studies on parenting and children’s obesity-related behaviors, especially nutrition behavior [38]. This possibly limits the generalizability of the results.
Another limitation possibly affecting generalizability of the results might be the slight overrepresentation of girls in the sample. However, in the study on the association between the FNC and adolescents’ consumption of healthy foods the correlations did not differ significantly between boys and girls [19].
Reviewer 2 Report
The authors have presented an extremely well written and sound manuscript with good attention to detail on the reporting and discussion of findings. It was a pleasure to read and very interesting in terms of the exploration of family dynamics on nutrition behaviors.
It should be possible to better highlight the use of the FNC in the Introduction section: e.g. how many studies have used it, is it widely used across geographic regions, are there any known limitations. This appears to be a very useful tool and would like to know more about it.
Minor language changes are suggested and better proofreading for precise language use, for example, replace in the Discussion section "However this does not entail" with "However this does not imply". Also, rewrite the sentence that begins with "This positive relation between the communication" to make it clearer and less complicated language for readers to better understand the meaning. Replace "thus not taking into consideration" with "thus not taken into consideration". Any other grammatical errors should be identified during copy editing by the journal or by the authors prior to publication.
Finally, in the implications section please state how the FNC may be utilized as a tool in future research, for other researchers across nutrition and behavior work, in addition to what the authors state where they "advocate studying general parental influences", what tools do they recommend to be used to study these?
Author Response
Dear reviewer,
Thank you for your compliments on the manuscript and the suggestions to improve the manuscript. Below we will provide a point-by-point response to your comments.
Reviewer comment 1:
“It should be possible to better highlight the use of the FNC in the Introduction section: e.g. how many studies have used it, is it widely used across geographic regions, are there any known limitations. This appears to be a very useful tool and would like to know more about it.”
Author’s response, comment 1:
Thank you for your suggestion. The FNC is a relatively new construct. Thus, not many studies have been conducted on the FNC. We aim to add to the knowledge concerning the FNC and its’ relation with children’s nutrition behavior. In the introduction, we added this information and explained the two studies that did use the FNC to study the relation between the FNC and individual’s nutrition behavior.
Corrections:
Introduction, line 84-92: The FNC is a relatively new construct and to our knowledge only two studies have been conducted using this validated instrument to measure the relation between the family and individual nutrition behavior [19, 20]. Both studies found a relation between the FNC and individual nutrition behavior: families with healthy behavioral patterns (consuming high levels of healthy nutrition and showing high levels of physical activity) rated their FNC higher compared to families with unhealthier behavioral patterns [20]. In the second study, it was found that a higher score on the FNC was related to a higher consumption of children’s healthy foods (i.e., fruit, vegetables, and salad) [19]. In both studies, the children were adolescents with a mean age of 14 years. It is unknown whether these results are similar for a population of younger children.
Reviewer comment 2:
“Minor language changes are suggested and better proofreading for precise language use, for example, replace in the Discussion section "However this does not entail" with "However this does not imply". Also, rewrite the sentence that begins with "This positive relation between the communication" to make it clearer and less complicated language for readers to better understand the meaning. Replace "thus not taking into consideration" with "thus not taken into consideration". Any other grammatical errors should be identified during copy editing by the journal or by the authors prior to publication.”
Author’s response, comment 2:
Thank you for the specific comment. The manuscript was sent to a professional editor (a native English speaker) before submitting the manuscript. However, we agree that some further changes could be made to improve the English and readability of the manuscript. Therefore, we read the manuscript thoroughly and corrected spelling mistakes. In addition, we rewrote some parts of the text to improve the readability of the manuscript. Our changes are marked in the manuscript. The main changes were done in the discussion (line 250 – 357). We also improved the sentences you suggested for improvement:
Discussion, line 290: However, this does not imply that the family influence is not present regarding the consumption of fruit.
Discussion, line 309: This positive relation between FNC-communication and children’s water consumption and between FNC-cohesion and children’s water consumption countered the negative relation between FNC-consensus and children’s water consumption.
Discussion, line 322: Another explanation may be that snacks and candies are not healthy foods and thus not taken into consideration when thinking about consuming healthy nutrition within the family.
Besides, we decided to talk about food and drink consumption or nutrition behavior instead of nutrition behaviors. Example of a correction made:
Results, line 224: FNC-Total correlated positively with healthy food consumption, i.e., fruit and vegetable intake, and negatively with unhealthy food and drink consumption, i.e., candy intake and soda consumption (Table 2).
Reviewer comment 3:
“Finally, in the implications section please state how the FNC may be utilized as a tool in future research, for other researchers across nutrition and behavior work, in addition to what the authors state where they "advocate studying general parental influences", what tools do they recommend to be used to study these?”
Author’s response, comment 3:
Again, we thank you for the suggestion. We decided to elaborate on the use of the FNC to measure the healthy family environment in the ‘Implications and recommendations’-section. Additionally, as suggested, we make some recommendations concerning tools to measure general parenting. We are aware of the fact that there are numerous tools to measure general parenting and it difficult to decide which tool best to use. However, because of the comprehensiveness of the Comprehensive General Parenting Questionnaire we recommend this questionnaire to measure general parenting. This questionnaire assess the five dimensions of parenting, whereas most other measures assess limited aspects of parenting. Due to the disagreement concerning the fit between food parenting practices concepts and constructs, we cannot recommend a specific tool to measure food parenting practices. We have made the following corrections in the ‘Implications and recommendations’-section:
Discussion, line 385-409: Our results add to the existing knowledge of the family influence on children’s nutrition behavior. The results underline the importance of addressing the whole family system instead of focusing merely on the parent-child subsystem [15, 40]. Addressing the family environment should be done by involving all family members [19]. The healthy family environment can be assessed by the individual evaluations of the FNC. Ideally, the FNC is measured by assessing the views of all family members (the parents, the child, and siblings) [18].
To inform intervention developers, we recommend further research into the interaction between the more proximal parent-level influences (e.g., parenting practices) and the more distal family-level factors (e.g., the FNC) and their interacting influence on the children’s nutrition behavior. To be able to study these interacting influences, we advocate studying general parental influences, i.e., general parenting, and more specific parenting, i.e., nutrition parenting practices, as well as the broader family context, i.e., the family nutrition climate.
To measure general parenting, we recommend the use of the validated Comprehensive General Parenting Questionnaire for caregivers of 5-13 year old children of Sleddens et al. [41]. This questionnaire was developed after a thorough search of the literature and assesses five parenting constructs: nurturance, structure, behavioral control, overprotection and coercive control [41]. Unfortunately, there is little consensus on how to best measure food parenting practices [42]. There are at least 71 instruments measuring food parenting practices [43]. In most cases, these instruments measure a small part of the spectrum of food parenting practices [43]. Fortunately, a comprehensive food parenting practices item bank is currently being developed which will allow a more consistent and comprehensive measurement of food parenting practices in the future [42]. For the measurement of the family-level influence, we recommend the use of the FNC.